# Nutritional Orthopedics and Space Nutrition as Two Sides of the Same Coin: A Scoping Review

**DOI:** 10.3390/nu13020483

**Published:** 2021-02-01

**Authors:** Matteo Briguglio

**Affiliations:** IRCCS Orthopedic Institute Galeazzi, Scientific Direction, Via Riccardo Galeazzi 4, 20161 Milano, Italy; matteo.briguglio@grupposandonato.it

**Keywords:** healthy eating, dietary supplement, musculoskeletal physiological phenomena, bones and bone tissue, sarcopenia, age-related bone losses, space travel, gravity, altered, nutritional physiological phenomena, aging prematurely

## Abstract

Since the Moon landing, nutritional research has been charged with the task of guaranteeing human health in space. In addition, nutrition applied to Orthopedics has developed in recent years, driven by the need to improve the efficiency of the treatment path by enhancing the recovery after surgery. As a result, nutritional sciences have specialized into two distinct fields of research: Nutritional Orthopedics and Space Nutrition. The former primarily deals with the nutritional requirements of old patients in hospitals, whereas the latter focuses on the varied food challenges of space travelers heading to deep space. Although they may seem disconnected, they both investigate similar nutritional issues. This scoping review shows what these two disciplines have in common, highlighting the mutual features between (1) pre-operative vs. pre-launch nutritional programs, (2) hospital-based vs. space station nutritional issues, and (3) post-discharge vs. deep space nutritional resilience. PubMed and Google Scholar were used to collect documents published from 1950 to 2020, from which 44 references were selected on Nutritional Orthopedics and 44 on Space Nutrition. Both the orthopedic patient and the astronaut were found to suffer from food insecurity, malnutrition, musculoskeletal involution, flavor/pleasure issues, fluid shifts, metabolic stresses, and isolation/confinement. Both fields of research aid the planning of demand-driven food systems and advanced nutritional approaches, like tailored diets with nutrients of interest (e.g., vitamin D and calcium). The nutritional features of orthopedic patients on Earth and of astronauts in space are undeniably related. Consequently, it is important to initiate close collaborations between orthopedic nutritionists and space experts, with the musculoskeletal-related dedications playing as common fuel.

## 1. Introduction

Over the centuries, the exploration of planet Earth required humans to venture into previously unknown territories. Food security in uncharted lands was of vital importance, and travelers had to establish agricultural or animal food systems as soon as supplies were running low [1]. This liaison between the environment and food resources is part of the nutritional paradigm that has always accompanied human survival on Earth: Environment ↔ Husbandry ↔ Food ↔ Health. Namely, the environmental conditions shape the procurement (animal or plant sources), the resulting food, and the nourishment of humans. If the *Environment* factors change, *Husbandry* and *Food* can adjust to balance human *Health*. The connection between *Food* and *Health* is of critical importance as the former has been considered not only a source of energy but also a medicine for humans, long before the first great explorations. For instance, Hellenic soldiers consumed high-iron foods to gain stamina before battles and village doctors recommended diverse dietary supplements, such as watercress or garlic, for enhancing recovery [2,3,4]. The nutritional paradigm has remained the same up to the present day: hostile surroundings need food systems that support good health, with some exceptions nevertheless (e.g., nutritional transition [5]: ↓environmental hostility, ↑engineering of husbandry, ↑food security, but ↓health). In this context, hospitals and space are normally considered unfavorable environments, with diminished food and health circumstances. Indeed, nutrition sciences have specialized in each research context to compensate for these inadequacies, thus giving birth to the sciences of Nutritional Orthopedics and Space Nutrition.

The field of nutrition applied to Orthopedics has developed in recent years, driven by the need to improve the efficiency of the treatment path by enhancing the recovery after surgery [6,7]. One of the main complexities of the field is the fact that orthopedic surgery (e.g., knee/hip replacement and spine surgery) finds its typical candidate to be an aged individual commonly suffering from osteoporosis and cardiovascular disease [8]. The hospital *Environment* further exposes the patient to stress, prolonged bed rest, and poor nutrition [9]. *Food* interventions to counteract bone and muscle loss or avert distinct nutrient deficits are, therefore, accessible (e.g., tailored diets, dietary supplements) [10]. Conversely, *Food* is almost considered an extra during pioneering space missions [11,12], because the human body acclimatizes fairly well to the short-term flights (i.e., near-Earth explorations, within 120 miles above Earth [13]). However, longer voyages (i.e., interplanetary or orbital explorations, with trips lasting from weeks to years) heading to deep space expose humans to an insecure food system, osteosarcopenia, and malnutrition, all leading to difficulties in the rehabilitation after landing [14]. At zero gravity, the *Environment* factor directly affects *Food* and *Health*, the liquids break into small drops, solid foods tend to form dust (e.g., original food for space flights included bite-sized cubes and dense liquids stuffed in tubes), the body floats, and gastro-intestinal function is close to being dysfunctional [15]. Advanced food systems are to be planned for space, thus avoiding current feedback loops on Earth that lead to pollution, destruction of habitats, natural resources, and loss of biodiversity [16]). At the time when we aspire to send people to distant planets, space food research has never been more important.

Some of the above-mentioned information already anticipates some similarities between the two fields of research, orthopedic patient vs. astronaut, hospital vs. space. However, it is indistinct in the existing literature about what kind of situations orthopedic patients and astronauts share, suffer from, and what similar hazards undermine their physical and mental health. To describe these issues, a scoping review was conducted to examine the nature of the interests and evidence common to Nutritional Orthopedics and Space Nutrition, both grounded on the millennia-old nutritional paradigm mentioned above. Specifically, the question to be answered is “What is known about the commonalities between Nutritional Orthopedics and Space Nutrition?”. The query was addressed following the subdivision of each research area into parallel phases: (1) pre-operative vs. pre-launch nutrition, (2) hospital-based vs. space station nutritional issues, and (3) post-discharge vs. deep space nutritional resilience. The existing knowledge guiding the path of care of orthopedic patients before/after surgery was mapped in parallel to specular phases of space research, being the pre-launch programs and nutritional support for short/long-lasting stay in space. The description of environmental hazards and illnesses was also part of the scoping review.

## 2. Methods

The review was conducted using the PRISMA−ScR (Preferred Reporting Items for Systematic Reviews and Meta-Analyses−Extension for Scoping Reviews; Appendix A; equator-network.org/reporting-guidelines/prisma-scr/), with no review protocol being currently available. Criteria for document inclusion comprised: the focus on orthopedic surgery or space travel, the discussion of nutritional issues, the development of the topic according to health-related frameworks for human survival, year of publication between 1950 and 2020, and text in English. Online sources covered PubMed and Google Scholar through a research strategy reported in Appendix B (Figure A1**)** (date of the most recent search: 22 September 2020). Identified terms were changed appropriately from PubMed to Google Scholar databases. Final documents were exported and duplicates were removed. The records were integrated by searching authoritative online sources (e.g., ntrs.nasa.gov). The process for selecting the evidence sources was conducted by a single experienced librarian who screened, charted, and abstracted independently. To ensure relevance to nutritional sciences for orthopedic (old) patients or astronauts, the primary selection focused on the words in titles/abstracts and then full texts, with the consideration of any type of document, including but not limited to animal/human studies, non-systematic and systematic reviews, meta-analyses, expert opinions, brief research/technical reports, commentaries, and press releases. This scoping review focused on the broad commonalities and, therefore, no specific type of article was excluded from the selection process. Eligible records were reported according to the year of study, study design, population, findings, and data were examined according to the relevant information useful to answer the study query. Specifically, the following data were extracted (orthopedic surgery vs. space travel): the existence and nature of nutritional support programs, the existing/desirable food systems, the existence and nature of environmental hazards, the need for particular nutrients that guarantee nutritional resilience, and the existence and nature of future research directions. No critical appraisal was conducted. To answer the review question, the range of evidence was grouped and narratively synthesized according to the following sections: nutritional needs and interventions before elective orthopedic surgery (home-based) vs. before space launch (Earth-based); the food procurement systems in hospitals vs. in space; research directions in the field of food science and nutrition in hospitals vs. in space; the hazards to human survival in a hospital vs. in space; the nutritional issues after hospitals vs. space deconditioning. Significant information on Nutritional Orthopedics and Space Nutrition is visually represented in Appendix D (Figure A3) and Appendix E (Figure A4), respectively.

## 3. Results

After the online search, a total of 44 documents (42 peer-reviewed articles + 2 book chapters [17,18]) on Nutritional Orthopedics and 44 documents (37 peer-reviewed articles + 6 documents ([19,20,21,22,23,24]) from the National Aeronautics and Space Administration + 1 document [25] from the Federal Aviation Administration (FAA)) on Space Nutrition were selected (see Appendix C (Figure A2) for the flow diagram). The scoped nutritional commonalities between the two fields of research, with relevance to the specific areas of nutritional interest mentioned above, were narratively described in the next passages (see Appendix F (Table A1)). Further details of all documents are reported in Appendix G (Table A2).

### 3.1. Pre-Operative (Home-Based) vs. Pre-Launch (Earth-Based) Nutritional Issues

#### 3.1.1. Nutritional Program Prior to Elective Orthopedic Surgery

In the mid-1990s, a state of malnutrition was already recognized as a risk for negative outcomes in surgical patients [26]. Malnutrition in Orthopedics may be detected through biochemical, anthropometric, functional, or dietetic assessments [10]. Nonetheless, it is sometimes preferred to adopt a population medicine approach in which all patients undergo nutritional optimization, regardless of their nutritional status. In fact, some age-related comorbidities and medications could inevitably worsen micronutrient reservoirs during the period up to surgery, even in the presence of a good nutritional status at the evaluation prior to admission [27]. Some patients may necessitate different routes of nutrition administration, depending on the ability to chew or on the gastrointestinal function. Enteral, parenteral, or modified texture foods are considered valuable options to attain the best possible condition before surgery.

Contrariwise, the significance of food abstention immediately before surgery is one of the most debated tenets in the field of pre-operative nutritional programs. On the one hand, there appears to be no general association with vomiting or aspiration but, on the other hand, there is a lack of available data on individual surgical procedures [28,29]. Beyond the habit of draw indications from parallel extrapolations, replacing the absolute fasting from midnight with a calorie-loaded supplement up to a few hours before surgery may be a safe element for perioperative care as far as Orthopedics is concerned [30,31].

Some deficiency syndromes, such as iron deficiency and hypovitaminosis D, are a very common burden in aged orthopedic patients and engineered formulas that facilitate intestinal absorption are therefore used (e.g., oil-in-water emulsion with propylene glycol for vitamin D [8] and sucrosomial matrix for iron [27]). Proper perioperative interventions should comprise diet and supplements tailored according to age, sex, and disease conditions, integrated with physical training, psychological support, smoking cessation, and alcohol reduction [32,33,34]. In orthopedic patients, the body composition should not only favor lean mass formation, which is vital for muscle and bone components, but also guarantee adequate fat reserves that harbor a source of energy, protect bones from traumas, and store lipophilic vitamins [35,36]. Some micronutrients, such as vitamin A and E, play major roles in the proper functioning of the immune system [37], and deficiencies should be corrected especially in aged individuals who are ordinarily affected by immuno-senescence and wound healing defects [38].

#### 3.1.2. Nutritional Conditioning Prior to Launch and “Packed Launch”

In the mid-1900s, it was already known that the astronauts must be fit to travel [39]. After all, “Finisque ab origine pendet” (Astronomica of Mark Manilius—first century C.E.). Therefore, regardless of the length of the expedition, space travelers should be subjected to nutritional optimization before departure. The goal is to achieve good body reservoirs, to avoid deficiencies of vitamins or minerals, and to obtain nutrition-derived immunocompetence from tailored supplements, such as omega-3 fatty acids [40]. Tailored dietary prescriptions and intake monitoring start months before launch, with astronauts being able to select their own space menu [19]. This personalization is known to guarantee the maximization of both physical and mental status [41].

Space travelers leave behind earthly cravings that are either nutritionally poor or counterproductive, such as junk foods, sodas, or alcoholic beverages. For example, carbon dioxide bubbles do not float in microgravity and remain in the beverage even after drinking, forming foam in the intestine, whereas ethyl alcohol is known to increase susceptibility to disorientation, to reduce reaction time and tolerance to g-forces, to impair physical performance and mental judgment [25]. When the launch day arrives, space travelers should refrain from eating food immediately prior to launch as the thrusting into microgravity is known to cause stomach awareness, nausea, and vomiting [20].

Space food should be mostly freeze-dried in individual packages to reduce water activity and help consumption. Furthermore, each package must include an adhesion system, like Velcro, to prevent it from floating. Space food should also be compact in order to take up little space (cargo capacity in space flights is limited) and to prevent dispersion due to microgravity once the package is opened [42]. The surface tension that forms with rehydration is useful to prevent the crumbs from flaking. Overall, the ultimate success of a space mission relies on food safety and stability, which collectively refer to the long-term preservation of microbiological, sensory (color, taste, flavor, and texture), and nutritional characteristics [42]. Historically, inadequate provisions are known to lead to ill-fated expeditions [43]. Since it is not feasible to overload the spaceship with food supplies and it is not convenient to keep resupplying the crew beyond near-orbit stations [44], astronauts will have to start growing their own food.

Summary I. A total of 13 references ([8,10,19,20,25,27,29,33,34,38,39,40,42]) were used to scope the following nutritional commonalities between pre-operative (home-based) vs. pre-launch (Earth-based) issues. The relevance of having good nutritional status; the importance to acquire nutrition-derived immunocompetence; the significance of food abstention before stressful situations; the goal of giving up/reducing bad habits; the need to opt for tailored nutrition and advanced dietary preparations.

### 3.2. Post-Operative (Hospital-Based) vs. Space Station (Planetary) Nutritional Issues

#### 3.2.1. Hospital Food Systems: Every Day Your Food Tray

Hospitals can adopt two main types of foodservice models according to clinical and organizational measures [45]:“Cook and Chill Systems”: outsourced catering systems that prepare, pack, and ship single-serving meals to the hospital, which is therefore equipped with a minimal-functionality kitchen.“Cook and Serve Systems”: in-house fully operational kitchen that processes and serves the meals directly to the wards.

A mixed model, with both precooked and fresh food, involves the meal transport to the ward and its cooking by using microwave ovens, allowing patients to order meals up to two hours prior to meal-times [46]. In the case of in-house kitchens, the staff provides food to different consumers, such as patients and employees, and, therefore, high amounts of food are produced simultaneously by sharing resources and equipment. However, this model is preferable because it allows better management, for example, of last-minute orders. At the same time, the kitchen should be well organized with separated compartments for storage in different environments and with spacious areas to avoid cross-contamination. Meals brought from the kitchen must be delivered on separate trays in packaging suitable for avoiding the partitioning of chemical compounds into food [47]. The patient should be able to wash his hands before meal consumption and food has to be at the right temperature. The control of the process should be as far upstream in the system as possible through continuous monitoring techniques. The modern food risk analysis and control procedures were pioneered in the 1960s for the production of safe foods for the United States space program [48]. Additionally, it should be considered that a meal’s value also depends on the quality of raw materials concerning sensory characteristics, purity, contamination, radioactivity, adulteration, and loss on drying [49]. The engagement between the patient and the meal order staff should be improved through a bedside spoken meal ordering system [50], being the productivity of the entire foodservice model primarily relied at the point of food consumption. In fact, plate waste is the major determinant of foodservice model losses [51].

#### 3.2.2. Food Research Laboratory in Hospital

Special diets for food intolerances, allergies, nutrition-related diseases, and dietary/religious dogmas should be handled by the clinical nutrition unit, working in parallel with the research laboratory, with the aim of evaluating the effectiveness of dietary interventions. Clinical- and patient-oriented outcomes should complement the “what, when, and how”, the decision-making algorithms, and the cost analysis [10,52]. Several research objectives are worth pursuing. The obstacles to early feeding such as the motion sickness exacerbating nausea/vomiting, should be investigated, considering that a diverse combination of medications might be associated with different symptoms [7]. Patients suffering from osteosarcopenia may benefit from calcium and protein supplements [53], preparations of vitamin D and iron can optimize cardiac function and iron status [27,54], and the Mediterranean diet may slow down the age-related joint degeneration and subchondral bone deterioration [55,56]. Food preservation methods are crucial for food safety and quality. Heat remains the most widely used inactivation technique, but pressure or magnetic/electric field applications are also promising [57,58]. These operations should be adjusted according to performance standards (e.g., microbial inactivation, retention of vitamins). Older orthopedic patients should be presented with appetizing meals that stimulate the age-related decay of smell, taste, and hedonic appreciations [59,60]. Ready-to-use flavor enhancers may be used to improve intakes [61], possibly reducing plate waste. Food surplus may also be associated with other causes nevertheless, such as poor meal quality and early satiety [62]. Overall, the collaboration among the foodservice, the clinical nutrition unit, and the research laboratory is certainly fundamental in order to guarantee a real benefit to hospitalized patients [63].

#### 3.2.3. Space Food Systems: Where the Light Source Is, the “Onion Spur” Grows

Any habitable space food system should be adapted to environmental conditions. Currently, two options are on the table [64].

“Transit Food Systems”: on-orbit food systems in microgravity are based on groundbreaking operations that include prepackaged supplies and low-volume storage.“Planetary Surface Food Systems”: partial gravity systems, such as those that are built on other planets, sustain Earth-like processes and include plant crops and animal farms.

Low gravity shapes the turgor of plant structures, altering the physiological responses. However, long-term persistence in space can generate new cultivars and ecotypes that are more resistant, as well as new subspecies and varieties with morphological advantages [65]. Common space-related constraints for aeroponic/hydroponic crops are the proper running of irrigation systems and nutrient pumping, air regeneration, water recycling, and the cultivation area [66,67]. All aspects should be integrated by innovative systems (e.g., the Passive Orbital Nutrient Delivery System or PONDS [21]). Light-emitting diodes replace the sunlight [68], as direct solar flares from the Orion Arm, along with other galactic radiations, are known to cause severe consequences, like DNA breaks, chromosomal aberrations, genome instability, and carcinogenesis [69]. Any animal husbandries are influenced by component choices analogous to vegetable crops. Naturally, the quality of the animal product depends on the nutritional content of the feed [70]. Regrettably, the nutritional loss from this transformation remains one of the most important factors on Earth that worsens system efficiency [71]. In the 1980s, insects were suggested as valuable food for space cooking [22], in alternative to large animals (to be excluded due to their excessive consumption of resources). Insect farms take up little space, provide a steady stream of high-quality proteins, and allow efficient conversion of waste into food. For instance, the black soldier fly is known to considerably reduce the environmental impact of food production [72]. Microgravity could create some problems to flying insects, making the larvae the primary choice [73].

#### 3.2.4. Food Research Laboratory in Space

Planetary stations are likely to benefit from the presence of food research laboratories. The broad spectrum of food systems will be investigated through advanced technologies for compact food stowage, analyses for quality assurance, heat sealing and vacuum packaging for long-term shelf life, and menu design for a variety of food choices and nutritionally complete plans [23,64]. Anticipated recycling should balance surplus disposal, without leading to the abovementioned environmental degradation observed on Earth. Current examples of research centers to solve the challenges of food in space comprise the Kennedy Space Center (NASA) aiming for the exploration of Mars [74] or the Space FoodSphere (Japan Aerospace Exploration Agency or JAXA) aiming for the exploration of the Moon [75]. Cutting-edge research aims to find new plant species that do not encounter microgravity- or radiation-derived inefficiencies like lettuce [76] and duckweed [77], or biofortification techniques through cellular agriculture methods to enrich space food with molecules of interest [78]. The quantity of certain nutrients, such as vitamin K and C, may be low before packing or are known to degrade to inadequate levels over time [79]. Consequently, nutritional stability should be required to enable long-term missions. The biological role of diverse nutritional, nutraceutical, and aromatic iotas is worthy of being studied, considering that beneficial components can be incorporated directly into new chemotypes [2,80,81]. For instance, space travelers will need extra calcium and vitamin D for bone weakening, the proper amount of proteins to counteract sarcopenia, low iron to avoid overload, and low-sulfur protein intake for acid-base balancing [54,82,83,84,85]. Extra flavor molecules to increase food enjoyment may be necessary because astronauts may experience a dulling of the olfaction that is known to reduce food intake [86]. This is probably due to nasal congestion (the bodily fluids move to the upper body [20,87]) or to the fact that aromas diffuse differently in the space environment. Novel packaging techniques should fulfill the needs of food preservation, sustainable use of space resources, and strategical area assignments (e.g., the area for meal socialization during homesickness) [67,88,89]. Altogether, the integrated activities among these facilities certainly lay the foundations for a successful mission into deep space.

Summary II. A total of 33 references ([7,10,20,21,22,23,27,45,46,50,51,52,53,54,55,56,59,60,61,62,63,64,66,67,68,73,76,77,79,82,83,84,85]) were used to scope the following nutritional commonalities between post-operative (hospital-based) vs. space station (planetary) issues. The use of single-serving prepackaged meals; the need to establish demand-driven systems that crosstalk and reduce food waste; the importance of guaranteeing food safety and human engagement throughout the process; the need to support the musculoskeletal health; the interest in some nutrients (i.e., proteins, calcium, iron, vitamin D); the need for boosting food flavors to increase appetite, palatability, hedonic appreciation, and food intake.

### 3.3. Hospital vs. Deep Space Hazards and Illnesses

#### 3.3.1. Bedrest, Acute Stress, and Isolation

Aging is intrinsically associated with dietary impoverishment, musculoskeletal involution, restlessness, and progressive reduction of the ability to cope with physical and psychological stress factors [90,91]. This reduced resilience tends to worsen during prolonged hospitalization, resulting in the so-called hospital-associated deconditioning [92,93]. The first studies on the effects of prolonged immobility were advanced by the United States space agency in the 1970s [94,95]. Currently, enhanced recovery protocols tend to be fast-tracks, thus emphasizing the short hospital stay compared to classic care pathways [10]. Many hospital-associated hazards are known to cause this hospital-associated deconditioning, including:Bedrest. When the body is lying down, upright valves of the vascular system do not minimize the gravity-associated fluid shifts. The plasma volume decreases and affects circulation [96], increasing the risk of syncope-related falls [97]. Prolonged immobility accelerates bone and muscle loss, exposing to osteosarcopenia and increased risk of fall-related fractures [97,98]. Immobility is also associated with gastrointestinal affections, such as gastroesophageal reflux, low appetite, slow peristaltic rate, and constipation [17].Acute stress. Surgical incision elicits both local (e.g., tissue injury, inflammation, neuroendocrine activation) and systemic (e.g., stress hormones, altered circadian entrainment) responses [18], leading to a wide range of pathological alterations, such as skeletal muscle protein degradation, illness-promoting dysbiosis, and behavioral/psychological disturbances [99]. Surgical pain and post-operative nausea are known to decrease appetite and nutrient intake, thus further delaying recovery [100,101,102].Isolation. The unfamiliar hospital diet and sleep-wake rhythm affect the older adults in hospitals, triggering new biological routines. Limited access to visitors, role changes, and relocation have been known for decades to increase loneliness [103]. The sensorial deprivation due to limited interactions outside of the room exposes the old man to delirium along with age-associated spatial disorientation, disequilibrium, and drug-aggravated cognitive decline [90,104].

#### 3.3.2. Microgravity, Radiations, and Confinement

The spiral structure of our galaxy (the Milky Way) contains stars, planets, gas, and dust held together by gravitational forces. Our body evolved into this constant gravitational force of 1 g that has been conditioning the musculoskeletal, cardiovascular, and neuro vestibular system, thus shaping our perception of open spaces and our own body [87,105,106]. During short-term space flights, most of the body’s tissues undergo regular accommodative processes [107]. However, the further away from Earth the more the body will be exposed to various hazards that could cause mental and physical disturbances, all being problematic from both a healthy and operational viewpoint [20,24].

Microgravity. The absence of eccentric forces exposes the body to the loss of muscle volume/strength and changes in both the composition of muscle fibers and the capillary network [108]. Along with neuromuscular deconditioning [109], the skeletal involution is the most significant alteration that arises after prolonged habitation in space, mainly deriving from the lack of both mechanical forces and sun exposure. Weightless-related conditions also affect gas exchanges and cause cardiovascular derangements, both being linked to fluid accumulation and irregular perfusion [20,110].Radiations. In outer space, it is not possible to grow plants in the sunlight. High-energy heavy-ion charged and solar energetic particles cause genome instabilities, carcinogenesis, tissue degeneration, and neurobehavioral disorders [111]. Unfortunately, radiation-derived oxidative stress is deleterious also for bone stability [112], further aggravating the involution of the skeleton.Confinement. Prolonged residence in the spaceship’s enclosed habitat exposes the astronauts to hypobaric/hypoxic situations [113] and cognitive decrements that undermine the wellbeing of the entire crew [114]. Space travelers will live in enclosed spaces with artificial illuminations, but crossing multiple time zones and exposure to different light emissions will change the circadian rhythm. The onset of mood swings and sleep disturbances can be unavoidable, being aggravated by the stressful duties [115].

Summary III. A total of 30 references ([17,18,20,24,87,90,91,92,93,95,96,97,98,99,100,101,102,103,104,105,106,107,108,109,110,111,112,113,114,115]) were used to scope the following nutritional commonalities between hospital vs. deep space hazards/illnesses. The interest in protecting the musculoskeletal system from reduced mechanical forces that expose the individual to an increased risk of fractures; the environmental consequences on the cardiovascular system; the relevance of stressful situations in causing metabolic and neurobehavioral disturbances; the potential consequences of prolonged periods of solitude.

### 3.4. Hospital-Associated vs. Space-Associated Deconditioning

#### 3.4.1. Nutritional Resilience in Orthopedic Patients

The aged orthopedic patient is a frail individual, and the hospital-associated hazards add fuel to the fire. The efficiency of the entire food system, which includes the nutritional support program, lies in the ability of the various services to communicate. If food requirements are miscalculated and nutritional sustenance is not provided, the patient’s discontent would be more likely associated with poor food intake and greater deconditioning [10]. The research laboratory should appropriately estimate energy-protein needs and the hospital food system should continuously deliver the planned food. The physiological consequences of geriatric anorexia, such as dysphagia and hyposmia [86,116], can be counteracted by food manipulation (e.g., modified texture, flavor improvement), feeding assistance, and endorsing conviviality. The excessive culinary expectations can thus be mediated. Supplements for osteosarcopenia may include diverse molecules of interest, such as β-hydroxy-β-methylbutyrate, vitamin D, calcium, and amino acids [53,117], without assuming the disconnection from other non-pharmacological interventions like daily range-of-motion exercises [118]. In the near future, most of the orthopedic care pathways will evolve, integrating on-site prescriptions with home-based monitoring after discharge, but the aforementioned hazards will persist at the patient’s home. Reduced food supply, undermined food security, and the lack of desire to feed can destabilize the nutritional status of community-dwelling lonely individuals [37]. A balanced and appetizing diet, proper meal timing, and the avoidance of foods interfering with medicines should be addressed through telemedicine interventions [90,119]. It is of paramount importance to overcome a pre-existing/hospital-derived osteosarcopenia, as it is known to be associated with poor outcome after surgery and increased risk of new fall-related traumas [120]. Undesirably, there remains uncertainty on how to defy this decay [121], and (with current knowledge) the further we move away from the peak of bone mass the more we are destined to fall.

#### 3.4.2. Nutritional Resilience in Astronauts

Many of the degenerative processes in space have many common features with the aging process on Earth [122,123], including changes in body composition, osteosarcopenia, the risk of osteoporotic fractures, nutritional frailty, hypophagia, flavor/pleasure issues, cardiovascular diseases, and neuropsychiatric derangements. As future missions will last for years, being self-sufficient is critical and so is the efficiency of the food system. Space travelers will be able to maintain good nutritional status, overcome the physiological and psychological consequences caused by a long stay in space, providing top performance every day in the most hostile environments. Considering food-interaction design, space travelers will also have to produce/process/pack their own food through novel strategies, actualizing original experiences of multisensory eating (e.g., Flavor Journey 3D Printer) [124]. The daily dietary program should meet energy and nutrient requirements, filling the deficits or reducing the intake of counterproductive substances, such as the aforementioned acidifying proteins [125]. Planning of meal timing is of great importance to regulate the sleep-wake cycle, and it can prevent the unsettling of food security or time allotted for meals [125,126]. The electrolyte imbalances and changes in splanchnic blood flow are known to be associated with delayed gastric emptying and intestinal transit [127,128], easily leading to constipation, poor assimilation of nutrients, disruption of the epithelial barrier [129], and altered gut microbiota [130]. It is, therefore, necessary to ensure a daily balance of fibers and liquids, with the possible addition of bioactive substances like caffeine, which is recommended for intense performance on Earth [131]. Nevertheless, such substances are known to interfere with critical pharmacological pathways [132] and susceptible to adverse reactions [133,134]. Defying the depletion of body reserves is also of utmost importance to prevent osteoporosis-related fractures during the transition from a low gravity field to hypergravity, which happens in the conceivable landing on unexplored planets (breaking the hip while landing on an alien planet is not recommended). It is reasonable to believe that the better the nutritional status before departure, the easier and longer the space traveler will endure the environmental conditions of space. Unfortunately, we should admit that (with current knowledge) the closer to alien worlds, the weaker the bones will be.

Summary IV. A total of 20 references ([10,33,53,90,116,117,118,119,120,121,122,123,124,125,127,128,129,130,133,134]) were used to scope the following nutritional commonalities between hospital-associated vs. space-associated deconditioning/resilience. The significance of having an efficient food system that contributes to the maintenance of the individual’s good nutritional status; the goal of reducing nutritional deficits and the intake of counterproductive substances; the importance of meal timing; the need to balance the musculoskeletal involution; the goal to avoid the recurrence of osteosarcopenia-related traumas.

## 4. Discussion

The nutritional paradigm has played a major role in human evolution since the beginning, with the *Food* factor being considered daily nourishment, sustenance during migrations, ration during explorations, fuel during skirmishes, or medicine for recovery. Any environment requires adaptation, and similar ecological hazards are known to expose humans to comparable risks and consequences. The science of Nutritional Orthopedics and Space Nutrition have individually investigated for years similar hostile environments, dealing with the aging orthopedic patient on Earth and the degenerative processes due to prolonged stay in space, respectively. Beyond the authoritative reviews on each topic that convey fundamental details [10,14,67], the purpose of this article was to show the broad spectra of the nutritional aspects that Orthopedics and Space Research have in common, focusing on the extent rather than detailed evidence. A total of 88 documents published in the last 70 years were identified, thus showing that each research field has much to share with the other. The most relevant nutrition-related matches are:The environmental risks, such as food insecurity, reduced movement, and stressful situations.The negative consequences, comprising malnutrition, osteosarcopenia, and low food intake.The three control phases of optimization, food security, and resilience.The basic strategies of a balanced diet, supplementation, and engineered food systems.

This scoping review has some limitations. First, the nature of the question may have excluded data on nutritional aspects peculiar to a single field, as unshared information was not reported. Furthermore, the research methodology may have overlooked the positive aspect of data charting among more than one reviewer, which would have ensured the accuracy of the collected data. However, the strength of this scoping review lies in the parallel mapping of two fields of research in nutrition never presented before. This method of organizing a nutritional support program into three different phases (optimization, acute phase, and resilience) could be a vital element that simplifies the comparison between any apparently diverse and complex setting. This certainly advocates not only for the need for high-quality research in each area but also for the feasibility of translating scientific evidence into different disciplines. An alignment of the shreds of evidence from Nutritional Orthopedic and Space Nutrition seems to be reasonable for mutual enrichment, with the musculoskeletal-related studies playing as communicating vessels. Back in 1959, Space Nutrition was introduced with these words: “…the scientist concerned with research in nutrition is about to be charged with the task of solving man’s nutrition problems in the milieu of space before he has more than scratched the surface in solving the problems of man in the environment of Earth” [15]. Going into space allowed us to look at our planet from another perspective. Discoveries from space researches provided broader perspectives and physiological insights to enhance different aspects of Earth-based healthcare [87], such as the abovementioned translation of hazard analysis or bed-rest studies in food industries or gerontology, respectively. However, it is also possible to exploit the nexus of learning in the opposite direction and it is, therefore, reasonable to assume that space-based tasks can be deciphered by corresponding Earth-based operative solutions. Decades of knowledge are inherited in hospitals specialized in musculoskeletal diseases and in the industries operating in the field of food supplements, modified texture foods, active packaging, and flavor enhancers. Whether we are dealing with aged orthopedic patients or space travelers, a nutrition professional specialized in the musculoskeletal system may be the common handler able to drive two trains without derailing.

## Data Availability

Not applicable.

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
