# Peer review of "Nutritional Orthopedics and Space Nutrition as Two Sides of the Same Coin: A Scoping Review"

_nutrients, 2021, doi:10.3390/nu13020483_

Round 1
Reviewer 1 Report
Nutritional Orthopedics and Space Nutrition as two sides of 2 the same coin: a scoping review
Matteo Briguglio, Nutrients
The article by Matteo Briguglio is a revision of a previously submitted article discussing the overlap within the fields of nutritional orthopedics and space nutrition. In its present format, the article is much improved and has a logical flow that makes the article perspective meritorious. However, there is still a great deal of confusion for the reader due to overall formatting, and this confusion makes the connection between Nutritional Orthopedics and Space Nutrition asserted by the author unclear.
Lines 425-431 in the “Concluding Matches” section suggest that the article is meant to show the “…aspects shared between Orthopedics and Space Research,” and later asserts that the article “…[provides] that each research field has much to learn from each other.” However, each section dealing with Orthopedic nutrition still seems isolated from its corresponding section on space nutrition. The review would flow much better if direct parallels between nutritional orthopedics and space medicine were pointed out in each individual section. As it is currently written, however, the sections do not directly draw these parallels.
For example, section 4.2 discusses the hospital food research laboratory and simply discusses optimal strategies for food research within a hospital setting, without references to space travel. Section 4.4 then discusses the general expectations of the space food research laboratory, again without any references to parallels within a hospital setting. It seems that the parallels are alluded to (e.g. when flavor enhancers are discussed in both sections given loss of taste and smell associated with both space travel and aging). However, direct statements connecting the two fields are rarely made. These statements could be something like the example below:
“Similar to the age-related decline in taste and smell seen in post-operative orthopedic patients, space travelers experience olfactory dulling which is know to decrease food intake [82]”.
The article seems to have a complete lack of these connecting sentences and thus does not do a sufficient job of convincing the readers that the fields of Nutritional Orthopedics and Space Nutrition can benefit from each other. The article should thus be revised further to draw more direct attention to the overlap within the fields.
Author Response
Response to Reviewer 1 for manuscript nutrients-1077602
I would like to thank the Reviewer 1 for the comprehensive evaluation of the article. I believe I have answered the requests of all the reviewers. I now hope that the manuscript is worthy of being considered for publication. Below are the answers to your requests.
Corrections from Authors to Reviewer 1
- The article by Matteo Briguglio is a revision of a previously submitted article discussing the overlap within the fields of nutritional orthopedics and space nutrition. In its present format, the article is much improved and has a logical flow that makes the article perspective meritorious. However, there is still a great deal of confusion for the reader due to overall formatting, and this confusion makes the connection between Nutritional Orthopedics and Space Nutrition asserted by the author unclear. Lines 425-431 in the “Concluding Matches” section suggest that the article is meant to show the “…aspects shared between Orthopedics and Space Research,” and later asserts that the article “…[provides] that each research field has much to learn from each other.” However, each section dealing with Orthopedic nutrition still seems isolated from its corresponding section on space nutrition. The review would flow much better if direct parallels between nutritional orthopedics and space medicine were pointed out in each individual section. As it is currently written, however, the sections do not directly draw these parallels. For example, section 4.2 discusses the hospital food research laboratory and simply discusses optimal strategies for food research within a hospital setting, without references to space travel. Section 4.4 then discusses the general expectations of the space food research laboratory, again without any references to parallels within a hospital setting. It seems that the parallels are alluded to (e.g. when flavor enhancers are discussed in both sections given loss of taste and smell associated with both space travel and aging). However, direct statements connecting the two fields are rarely made. These statements could be something like the example below: “Similar to the age-related decline in taste and smell seen in post-operative orthopedic patients, space travelers experience olfactory dulling which is know to decrease food intake [82]”. The article seems to have a complete lack of these connecting sentences and thus does not do a sufficient job of convincing the readers that the fields of Nutritional Orthopedics and Space Nutrition can benefit from each other. The article should thus be revised further to draw more direct attention to the overlap within the fields.
The manuscript was previously submitted as a Perspective, but it has been criticized for its lack of reporting rigor and thus was resubmitted as a Scoping Review. I am glad that you now consider it meritorious.
Concerning the concluding sentence “…each research field has much to learn from each other…”, it is to be referred to the subsequent sentences that report the potential translation of knowledge (e.g. hazard analysis and bedrest studies). Furthermore, the potential for mutual enrichment is also underlined later when it is reported “…Decades of knowledge are inherited in hospitals specialized in musculoskeletal diseases and in the industries operating in the field of food supplements, modified texture foods, active packaging, and flavor enhancers…”. In reference to your comment, however, I believe it is possible to modify the sentence you quote as follows: “…thus showing that each research field have much to share with the other..…”
Concerning your comment about the lack of clear parallels, I partially disagree with your comment. The subdivision of the sections “…one topic vs. another…”, the schematization in point-by-case paragraphs and bulleted lists in bold, the clearly mirror images of Appendixes B and C, and the scoped features in Appendix D can be considered positively exhaustive. However, I understand the nature and suggestion of your comment and I therefore proceed to insert 4 new paragraphs at the end of sections 3, 4, 5, and 6 in order to emphasize the aspects that unite each phase of the process of Nutritional Orthopedics and Space Nutrition. The following paragraphs were added:
At the end of section 3. Pre-operative (home-based) vs. pre-launch (Earth-based) nutritional issues: “…Scoped nutritional commonalities between pre-operative (home-based) vs. pre-launch (Earth-based) issues: the relevance of having good nutritional status; the importance to acquire nutrition-derived immunocompetence; the significance of food abstention before stressful situations; the goal of giving up/reducing bad habits; the need to opt for tailored nutrition and advanced dietary preparations. …”.
At the end of section 4. Post-operative (hospital-based) vs. space station (planetary) nutritional issues: “…Scoped nutritional commonalities between post-operative (hospital-based) and space station (planetary) issues: the use of single-serving prepackaged meals; the need to establish demand-driven systems that crosstalk and reduce food waste; the importance of guaranteeing food safety and human engagement throughout the process; the need to support the musculoskeletal health; the interest in some nutrients (i.e. proteins , calcium, iron, vitamin D); the need of boosting food flavors to increase appetite, palatability, he-donic appreciation, and food intake. ….”.
At the end of section 5. Hospital vs. deep space hazards and illnesses: “…Scoped nutritional commonalities between hospital and deep space haz-ards/illnesses: the interest in protecting the musculoskeletal system from reduced me-chanical forces that expose the individual to an increased risk of fractures; the environ-mental consequences on the cardiovascular system; the relevance of stressful situations in causing metabolic and neurobehavioral disturbances; the potential consequences of pro-longed periods of solitude. ….”.
At the end of section 6. Hospital-associated vs. space-associated deconditioning: “…Scoped nutritional commonalities between hospital-associated and space-associated deconditioning/resilience: the significance of having an efficient food system that contributes to the maintenance of the individual’s good nutritional status; the goal of reducing nutritional deficits and the intake of counterproductive substances; the importance of meal timing; the need to balance the musculoskeletal involution; the goal to avoid the recurrence of osteosarcopenia-related traumas. ….”.

Reviewer 2 Report
The manuscript should follow all recommended PRISMA requirements. Please, include the following items in the methods section of your manuscript. If one item has not been completed during review, please state it and explain resons in your manuscript in order to provide reproducibility.
|
Protocol and registration |
|
Eligibility criteria |
|
Information sources |
|
Search |
|
Study selection |
|
Data collection process |
|
Data items |
|
Risk of bias in individual studies |
|
Summary measures |
|
Synthesis of results |
|
Risk of bias across studies |
|
Additional analyses |
According to these criteria, search strategy has been added, but the flow diagram was not provided and the checklist was not attached as a supplemental file for review.
Again, this manuscript could be of interest as a systematic review or following all PRISMA criteria, but this manuscript is not suitable for publication in its current form.
Author Response
Response to Reviewer 2 for manuscript nutrients-1077602
I thank the Reviewer 2 for the willingness to comment on my manuscript. I would like to point out that the past-resubmitted manuscript fully comprehended the checklist of Scoping Review PRISMA guidelines. I proceed to provide you with detailed responses to comments.
Corrections from Authors to Reviewer 2
The manuscript should follow all recommended PRISMA requirements. Please, include the following items in the methods section of your manuscript. If one item has not been completed during review, please state it and explain resons in your manuscript in order to provide reproducibility. Protocol and registration, Eligibility criteria, Information sources, Search, Study selection, Data collection process, Data items, Risk of bias in individual studies, Summary measures, Synthesis of results, Risk of bias across studies, Additional analyses. According to these criteria, search strategy has been added, but the flow diagram was not provided and the checklist was not attached as a supplemental file for review. Again, this manuscript could be of interest as a systematic review or following all PRISMA criteria, but this manuscript is not suitable for publication in its current form.
Thanks for the comment, with which I agree. Below is the detail of each ITEM that you can find added in the manuscript. The PRISMA extension for Scoping Reviews (PRISMA-ScR) of Andrea C. Tricco et al. (Annals of Internal Medicine, 2018 https://www.acpjournals.org/doi/10.7326/M18-0850) was followed. Newly introduced sections are the methodology of the review, the coupling of each phase in Orthopedics vs. the same area of Space Nutrition, the Appendix on the searching strategy, the Supplementary File of the selection of sources, and the table of synthesis.
Of note, the following items do not apply for ScR: Summary Measures, Risk of Bias, Additional Analyses.
ITEM 1 - Title: page 1.
ITEM 2 - Abstract: page 1.
ITEM 3 - Rationale: page 1-2.
ITEM 4 - Objectives: page 2. “ … This scoping review examined the nature of the interests and evidence common to Nutritional Orthopedics and Space Nutrition, both grounded on the millennia-old nutritional paradigm. Specifically, the question to be answered is “What is known about the commonalities between Nutritional Orthopedics and Space Nutrition?”. The query was addressed following the subdivision of each research area into parallel phases: 1) pre-operative vs. pre-launch nutrition, 2) hospital-based vs. space station nutritional is-sues, and 3) post-discharge vs. deep space nutritional resilience. The existing knowledge guiding the path of care of orthopedic patients before/after surgery was mapped in parallel to specular phases of space research, being the pre-launch programs and nutritional support for short/long-lasting stay in space. The description of environmental hazards and illnesses was also part of the scoping review..… ”
ITEM 5 - Protocol and registration, ITEM 6 - Eligibility criteria, ITEM 7 - Information sources: page 3. “…The review was conducted using the PRISMA−ScR (Preferred Reporting Items for Systematic reviews and Meta-Analyses−extension for Scoping Reviews; equa-tor-network.org/reporting-guidelines/prisma-scr/), with no review protocol being currently available. Criteria for document inclusion comprised: the focus on orthopedic surgery or space travel, the discussion of nutritional issues, the development of the topic according to health-related frameworks for human survival, year of publication between 1950 and 2020, publication in peer-reviewed journals or reporting in authoritative online sources (e.g. nasa.gov), and writing in English. Online sources covered PubMed and Google Scholar through a research strategy reported in Appendix A. The process for selecting the evidence sources included the evaluation of titles, abstracts and then full texts, leading to the charting of each document in one or more of the three stages mentioned above…”.
ITEM 8 - Search, ITEM 9 - Selection of sources of evidence†, ITEM 10 - Data charting process‡, ITEM 11 - Data items: page 11 & Appendix. “…a research strategy reported in Appendix A ……The online search strategy for PubMed (literature search started on September 22, 2020). Afterward, identified terms were changed appropriately for Google Scholar database. To ensure relevance to nutritional sciences for orthopedic (old) patients or astronauts, the primary selection focused on the words in titles/abstracts, with the consideration of any type of document, including but not limited, to animal/human studies, non-systematic and systematic reviews, meta-analyses, expert opinions, brief research/technical reports, commentaries, and press releases. This scoping review focuses on the broad commonalities and, therefore, no specific type of article was excluded from the selection process. The preference for citing one reference over another depended upon the authoritativeness of the publication. Specifically, the following data were abstracted (orthopedic surgery vs. space travel): the existence and nature of nutritional support programs, the existing/desirable food systems, the existence and nature of environmental hazards, the need for particular nutrients that guarantee nutritional resilience, the existence and nature of future research directions.…”.
ITEM 12 - Critical appraisal of individual sources of evidence§, ITEM 16 - Critical appraisal within sources of evidence: page 3, not relevant for the scope. “…The purpose of the review was to describe the existing nutritional knowledge and direction shared between Nutritional Orthopedics and Space Nutrition and no critical appraisal was consequently investigated. ….”.
ITEM 13 - Synthesis of results: page 3. “…To answer the review question, the range of evidence was grouped and narratively synthesized according to the following sections: nutritional needs and interventions be-fore elective orthopedic surgery (home-based) vs. before space launch (Earth-based); the food procurement systems in hospitals vs. in space; research directions in the field of food science and nutrition in hospitals vs. in space; the hazards to human survival in hospital vs. in space; the nutritional issues to counteract the deconditioning from hospitals vs. space..…”.
ITEM 14 - Selection of sources of evidence: page 3 & Supplementary. “…After the on-line search, a total of 46 documents (44 peer-reviewed articles + 2 book [95], [96]) were selected on Nutritional Orthopedics and 44 documents (37 peer-reviewed arti-cles + 6 documents from the National Aeronautics and Space Administration [32], [35], [61], [66], [69], [106] + 1 document from the Federal Aviation Administration [34]) were se-lected on Space Nutrition (see Supplementary for the flow diagram)..…”.
ITEM 15 - Characteristics of sources of evidence, ITEM 17 - Results of individual sources of evidence: page 14-17 (Appendix D).
ITEM 18 - Synthesis of results: page 3-9.
ITEM 19 - Summary of evidence: page 9-10 & Appendixes B and C. “…A total of 90 documents published in the last 70 years were identified that addressed the nutritional features that the orthopedic patient and the space traveler have in common (see Appendix D), thus showing that each research field have much to share with the other. The most relevant nutrition-related matches are: • The environmental risks, such as food insecurity, reduced movement, and stressful situations. • The negative consequences, comprising malnutrition, osteosarcopenia, and low food intake. • The three control phases of optimization, food security, and resilience. • The basic strategies of a balanced diet, supplementation, and engineered food systems.. ……………………………………………… Key aspects of human nutrition in orthopedic patients. The perioperative nutritional sup-port program for patients undergoing major orthopedic surgery comprises three phases. (I) First, the optimization of the nutritional status should indicate recommendations on healthy eating, tailored diets, and dietary supplements for nutrient deficits through a structured behavioral program. (II) Second, early oral feeding should be promoted to avoid fasting and to boost recovery. Hospital diets may be integrated according to the patient’s requirements, with services of food and beverages to patients following a model redesign to facilitate engagement and improve satisfaction. Case-specific prevention of food waste should be applied by promoting interaction in the order process and meal sharing. (III) Third, the variety of nutritional issues after discharge include not only high-risk pre-existing illnesses, like frailty, but also newly occurring conditions from hospital-associated deconditioning, like sarcopenia from immobility. Since the hospital could easily be considered a hostile Environment that affects Health of patients and limits the Husbandry factor, it is necessary to reach balance by acting on the central components of the nutritional paradigm, where Food really plays a central role.…………………………………………………………. Key aspects of human nutrition during space travel. The nutritional program aiming at supporting all the individual’s activities related to space travel should cover three main aspects. (I) First, the individual’s nutritional status should be optimized before launch. Environmental conditions, especially terrestrial gravity, are able to maximize psycho- and physiological interventions. (II) Second, an autonomous food system should be settled on space or planetary stations. Advanced plantations following the principles of eco-sustainability and precision nutrition might cohabit with farms of animals with a high feed conversion efficiency, such as insects. (III) Third, Space Nutrition research should focus on the issues about venturing into deep space. During prolonged absence of the gravitational system, the human body is subjected to various senile-like conditions, which are not necessarily caused by dietary factors, but that can be defied or at least delayed by integrated approaches comprising tailored nutrition. The nutritional paradigm initiated on Earth, where human beings have found a way to make the most of it, will undergo a drastic revolution when the Environment factor is the deep space. Since human Health is assumed to be constant, environmental deviations must be balanced with novel Husbandry systems and Foods…”.
ITEM 20 - Limitations: page 10 “ … This scoping review has some limitations. First, the nature of the question may have excluded data on nutritional aspects peculiar to a single field, as unshared information was not reported. Furthermore, the research methodology may have overlooked the positive aspect of data charting among more than one reviewer, which would have ensured the accuracy of the collected data. However, the strength of this scoping review lies in the parallel mapping of two fields of research in nutrition never presented before... ”
ITEM 21 - Conclusions: page 10. “…This method of organizing a nutritional support program into three different phases (optimization, acute phase, and resilience) could be a vital element that simplifies the comparison between any apparently diverse and complex setting. This certainly advocates not only for the need of high quality research in each area, but also for the feasibility of translating scientific evidence into different disciplines. An alignment of the shreds of evidence from Nutritional Orthopedic and Space Nutrition seems to be reasonable for mutual enrichment, with the musculoskeleton-related studies playing as communicating vessels.. …”.
ITEM 22 - Funding: page 10. “…This research received no external funding.…”.

Reviewer 3 Report
In this scoping review, Dr. Briguglio seeks to show the similar nutrition and health-related consequences, namely the deterioration of the musculoskeletal system, between orthopedic surgery and space travel.
A scoping review investigates a complex, heterogenous, or understudied topic to showcase the extent/range of the research topic, determine its value and potential scope, as well as summarize current research findings and identify any potential research gaps. A scoping review does not seek to provide a critical evaluation or recommendation, but to merely summarize and illustrate current findings.
Overall, Dr. Briguglio accomplishes this goal and provides a new perspective on two previously separated research domains. From this scoping review, this reviewer identifies both nutritional orthopedics and space nutrition as research fields set in hostile environments that require appropriate food systems to optimize health and recovery. Both nutritional orthopedics and space nutrition require 1) personalized pre-operation or pre-space flight nutritional optimization to maximize health, and both 2) an appropriate and readily available food supply and 3) specialized/nutrient-dense food sources to offset the deleterious effects of major surgery, bedrest, or low-gravity (namely osteosarcopenia and cardiovascular disorders). These similarities in both the type of nutritional intervention required and specific health outcomes of interests allows for the translation of research findings between nutritional orthopedics and space nutrition, and possibly mutual collaboration and enrichment.
Though Dr. Briguglio largely accomplishes the objective of the scoping review, improvements in paragraph organization, sentence structure, reading flow, and additional sub-paragraphing would improve readability and prevent miscommunication.
Overall, this reviewer recommends Dr. Briguglio connect with an additional reviewer to help improve paragraph structure (emphasizing a topic sentence structure), reading flow, and provide any additional gaps in the literature not identified by Dr. Briguglio.
Below are a few grammatical and organization areas which this reviewer believes need to be addressed.
Line 12: “Sundry” food challenges – Is this referring to the process of food liquid evaporation for space flight? If so, I do not believe all sources are “sundried”
Line 17: Change Online source to Pubmed & Google Scholar. Change ‘Last 70 years’ to specific dates ‘1950-2020’.
Line 35-37: Clarify and simplify statement. Odd choice of words
Line 43: …and Food “Can” change to assure balance
Line 64-68: Correct run-on sentence
Line 75-76: Add a “,” between field & nonetheless. “deep space exposes the human body to an insecure food system”
Line 80-81: “food systems have feedback loops which may negatively affect the planet and lead to pollution, destruction of habits, natural resources, and loss of biodiversity”
Line 81: “At zero gravity the Environment factor directly affects Food and Health, liquids break into small drops….”
Both Sections 3 & 4 would greatly benefit from better paragraph organization as seen in Sections 5 & 6. Creating sub-paragraphs instead of one single paragraph would improve readability and communicate more clearly.
Example organization and sub-paragraphing of section 3.2
Line 160-168: 1st sub-paragraph regarding nutritional optimization
Line 168-174: 2nd sub paragraph regarding micro-gravity food complications
Line 174-176: 3rd sub paragraph Day-of-Flight nutrition
Line 176-184: 4th sub paragraph Space food packaging
Line 184-188: 5th sub paragraph Historical context and complications.
Author Response
Response to Reviewer 3 for manuscript nutrients-1077602
I would like to thank Reviewer 3 for taking the time to revise my manuscript. I have proceeded to correct according to your suggestions. I hope the manuscript is now worthy of publication. Importantly, other than the revisions you requested, you will find some new items related to scoping reviews that Reviewer 2 has requested.
Corrections from Authors to Reviewer 3
- In this scoping review, Dr. Briguglio seeks to show the similar nutrition and health-related consequences, namely the deterioration of the musculoskeletal system, between orthopedic surgery and space travel. A scoping review investigates a complex, heterogenous, or understudied topic to showcase the extent/range of the research topic, determine its value and potential scope, as well as summarize current research findings and identify any potential research gaps. A scoping review does not seek to provide a critical evaluation or recommendation, but to merely summarize and illustrate current findings. Overall, Dr. Briguglio accomplishes this goal and provides a new perspective on two previously separated research domains. From this scoping review, this reviewer identifies both nutritional orthopedics and space nutrition as research fields set in hostile environments that require appropriate food systems to optimize health and recovery. Both nutritional orthopedics and space nutrition require 1) personalized pre-operation or pre-space flight nutritional optimization to maximize health, and both 2) an appropriate and readily available food supply and 3) specialized/nutrient-dense food sources to offset the deleterious effects of major surgery, bedrest, or low-gravity (namely osteosarcopenia and cardiovascular disorders). These similarities in both the type of nutritional intervention required and specific health outcomes of interests allows for the translation of research findings between nutritional orthopedics and space nutrition, and possibly mutual collaboration and enrichment. Though Dr. Briguglio largely accomplishes the objective of the scoping review, improvements in paragraph organization, sentence structure, reading flow, and additional sub-paragraphing would improve readability and prevent miscommunication.
Thanks for the words of support on my manuscript.
- Overall, this reviewer recommends Dr. Briguglio connect with an additional reviewer to help improve paragraph structure (emphasizing a topic sentence structure), reading flow, and provide any additional gaps in the literature not identified by Dr. Briguglio.
Thank you for the comment. I believe that the requests made by Reviewer 1 and Reviewer 2 can satisfy your demand.
- Below are a few grammatical and organization areas which this reviewer believes need to be addressed. Line 12: “Sundry” food challenges – Is this referring to the process of food liquid evaporation for space flight? If so, I do not believe all sources are “sundried”
I refer to the adjective “sundry” food challenges as a synonym of “divers” or “miscellaneous”. I substituted the word in the abstract with “varied”.
- Line 17: Change Online source to Pubmed & Google Scholar.
In the abstract, “…PubMed and Google Scholar were used to collect documents…”
- Change ‘Last 70 years’ to specific dates ‘1950-2020’.
In the abstract, “…documents published from 1950 to 2020…”
- Line 35-37: Clarify and simplify statement. Odd choice of words
In the section 1.1. The millennia-old nutritional paradigm on Earth, “…Over the centuries, the exploration of planet Earth required humans to venture into previously unknown territories. Food security in uncharted lands was of vital importance, and travelers had to settle agricultural or animal food systems as soon as supplies were running low …”
- Line 43: …and Food “Can” change to assure balance
I added the word “can”.
- Line 64-68: Correct run-on sentence
Thanks for the note. At the end of paragraph “1.2. Unravelling the paradigm on orthopedic patients: Nutritional Orthopedics”, I corrected the run-on sentence as follows. “…In the de-conceptualization of the abovementioned nutritional paradigm, years of nutritional researches have defined Food interventions to counteract bone and muscle loss or avert distinct nutrients’ deficits (e.g. tailored diets, dietary supplements) [10], thus re-calling ancient doctors’ approaches.…”
- Line 75-76: Add a “,” between field & nonetheless. “deep space exposes the human body to an insecure food system”
In the paragraph “1.3. Earth-to-space paradigm shift: Space Nutrition”, I corrected as follows. “…The human body acclimatizes fairly well to the short-term missions from a terrestrial to a low-gravity field. Conversely, longer voyages heading to deep space expose the human body to an insecure food system, malnutrition, osteosarcopenia and psychological disturbances, all leading to difficulties in the rehabilitation after landing …”
- Line 80-81: “food systems have feedback loops which may negatively affect the planet and lead to pollution, destruction of habits, natural resources, and loss of biodiversity”
In the paragraph “1.3. Earth-to-space paradigm shift: Space Nutrition”, I corrected as follows. “…Decisively, food systems have feedback loops that may negatively affect the planet and lead to pollution, destruction of habitats, natural resources, and loss of biodiversity …”
- Line 81: “At zero gravity the Environment factor directly affects Food and Health, liquids break into small drops….”
In the paragraph “1.3. Earth-to-space paradigm shift: Space Nutrition”, I corrected as follows. “…At zero gravity, the Environment factor directly affects Food and Health, the liquids break into small drops, solid foods tend to form dust (e.g. original food for space flights included bite-sized cubes and dense liquids stuffed in tubes), the body floats, and gastro-intestinal function is close to being dysfunctional …”
- Both Sections 3 & 4 would greatly benefit from better paragraph organization as seen in Sections 5 & 6. Creating sub-paragraphs instead of one single paragraph would improve readability and communicate more clearly. Example organization and sub-paragraphing of section 3.2 Line 160-168: 1st sub-paragraph regarding nutritional optimization. Line 168-174: 2nd sub paragraph regarding micro-gravity food complications. Line 174-176: 3rd sub paragraph Day-of-Flight nutrition. Line 176-184: 4th sub paragraph Space food packaging. Line 184-188: 5th sub paragraph Historical context and complications.
Thanks for the comment. While it is not possible to name the paragraphs of sections of 3.1 and 3.2 as bullet points lists below (of sections 4 and 5), I agree that the reorganization of the information would allow a more schematic reading. That is, I have reorganized the paragraphs 3.1 and 3.2, dividing the information into three point-to-line sections that reflect the nutritional preparation, the abstaining from food, and food details.
“3.1. Nutritional program prior to elective orthopedic surgery
In the mid-‘90s, a state of malnutrition was already recognized as a risk for negative outcomes in surgical patients [17]. Malnutrition in Orthopedics may be detected through biochemical, anthropometric, functional, or dietetic assessments [10]. Nonetheless, it is sometimes preferred to adopt a population medicine approach in which all patients undergo nutritional optimization, regardless of their nutritional status. In fact, some age-related comorbidities and …………………………..
Contrariwise, the significance of food abstention immediately before surgery is one of the most debated tenet in the field of pre-operative nutritional programs. On the one hand, it appears to be no general association with vomiting or aspiration but, on the other hand, there is a lack of available data on individual surgical procedures [19, 20]. Beyond the habit of draw indications from parallel extrapolations, replacing the absolute fasting from midnight with a calorie-loaded ………………………………….
Some deficiency syndromes, such as iron deficiency and hypovitaminosis D, are a very common burden in aged orthopedic patients and engineered formulas that facilitate intestinal absorption are therefore used (e.g. oil-in-water emulsion with propylene glycol for vitamin D [8] and sucrosomial matrix for iron [18]). Proper perioperative interventions should comprise diet and supplements both tailored according to age, sex, and disease conditions, thus being integrated with physical training, psychological support, smoking cessation, and alcohol reduction [23-25]. In orthopedic patients, …………………...”
“3.2. Nutritional conditioning prior to launch and “packed launch”
In the mid-1900s, it was already known that the astronauts must be fit to travel [30]. After all, “Finisque ab origine pendet” (Astronomica of Mark Manilius - first century C.E.). Therefore, regardless of the length of the expedition, space travelers should be subjected to a nutritional optimization before departure. The goal is to achieve good body reservoirs, to avoid deficiencies of vitamins or minerals, and to obtain nutrition-derived immunocom-petence from tailored supplements, such as omega-3 fatty acids [31]. Tailored dietary pre-scriptions and intake monitoring start months before launch, with astronauts being able to select their own space menu [32]. This personalization is known to guarantee the maximization of both physical and mental status [33].
Space travelers leave behind earthly cravings that are either nutritionally poor or counterproductive, such as junk foods, sodas, or alcoholic beverages. For example, carbon dioxide bubbles do not float in microgravity and remain in the beverage even after drink-ing, forming foam in the intestine, whereas ethyl alcohol is known to increase susceptibil-ity to disorientation, to reduce reaction time and tolerance to g-forces, to impair physical performance and mental judgment [34]. …………………
Space food should be mostly freeze-dried in individual packages to reduce water ac-tivity and help consumption. Furthermore, each package must include an adhesion sys-tem, like Velcro, to prevent it from floating. Space food should also be compact in order to take up little space (cargo capacity in space flights is limited) and to prevent dispersion due to microgravity once the package is opened [36]. The surface tension that forms with rehydration is useful to prevent the crumbs from flaking. Overall, the ultimate success of a space mission relies on food safety and …………………....…”
Moreover, in order to improve readability of the section 3, 4, and 5 and to communicate more clearly the parallels, I added 4 new paragraphs at the end of the respective sections.
At the end of section 3. Pre-operative (home-based) vs. pre-launch (Earth-based) nutritional issues: “…Scoped nutritional commonalities between pre-operative (home-based) vs. pre-launch (Earth-based) issues: the relevance of having good nutritional status; the importance to acquire nutrition-derived immunocompetence; the significance of food abstention before stressful situations; the goal of giving up/reducing bad habits; the need to opt for tailored nutrition and advanced dietary preparations. …”.
At the end of section 4. Post-operative (hospital-based) vs. space station (planetary) nutritional issues: “…Scoped nutritional commonalities between post-operative (hospital-based) and space station (planetary) issues: the use of single-serving prepackaged meals; the need to establish demand-driven systems that crosstalk and reduce food waste; the importance of guaranteeing food safety and human engagement throughout the process; the need to support the musculoskeletal health; the interest in some nutrients (i.e. proteins , calcium, iron, vitamin D); the need of boosting food flavors to increase appetite, palatability, he-donic appreciation, and food intake. ….”.
At the end of section 5. Hospital vs. deep space hazards and illnesses: “…Scoped nutritional commonalities between hospital and deep space haz-ards/illnesses: the interest in protecting the musculoskeletal system from reduced me-chanical forces that expose the individual to an increased risk of fractures; the environ-mental consequences on the cardiovascular system; the relevance of stressful situations in causing metabolic and neurobehavioral disturbances; the potential consequences of pro-longed periods of solitude. ….”.
At the end of section 6. Hospital-associated vs. space-associated deconditioning: “…Scoped nutritional commonalities between hospital-associated and space-associated deconditioning/resilience: the significance of having an efficient food system that contributes to the maintenance of the individual’s good nutritional status; the goal of reducing nutritional deficits and the intake of counterproductive substances; the importance of meal timing; the need to balance the musculoskeletal involution; the goal to avoid the recurrence of osteosarcopenia-related traumas. ….”.

Round 2
Reviewer 2 Report
Thanks for your response to my comments in your prior review. Nevertheles, some major flaws remain unclear:
1) Please, reorganize your manuscript according to the subsections of PRISMA criteria for methods section and in addition re-organize introduction, results, discusion and conclusion.
2) Methodology and results (Tables) for risk of bias evaluations have not been provided. In addition, methodology and results (Tables) for quality of studies have not been addressed.
3) The flow diagram of PRISMA criteria have not been added as a figure.
4) In addition, selected study data should be expanded in tables (impact factor of journals, more information on sample, studies findings...)
You have to re-organise your whole manuscript. Please, be accurate within PRISMA recommendations.
Author Response
Response to Reviewer 2 for manuscript nutrients- 1077602
- The manuscript has been reorganized in its titles and sections. The information requested by the PRISMA ITEMS has been moved to be part of the correct section of the checklist. All corrections are highlighted in light blue.
- The article, as it is now organized, adheres to the PRISMA standards for scoping reviews. The nature and purpose of the article cannot be translated into a systematic review. I would like to report these two extracts from the “PRISMA extension for Scoping Reviews (PRISMA-ScR) of Andrea C. Tricco et al. (Annals of Internal Medicine, 2018 https://www.acpjournals.org/doi/10.7326/M18-0850)”
"A key difference between scoping reviews and systematic reviews is that the former are generally conducted to provide an overview of the existing evidence regardless of methodological quality or risk of bias (4, 5). Therefore, the included sources of evidence are typically not critically appraised for scoping reviews."
"Item 15 (Not Applicable): Risk of Bias Across Studies This item from the original PRISMA is not applicable for scoping reviews because the scoping review method is not intended to be used to critically appraise (or appraise the risk of bias of) a cumulative body of evidence."
Therefore, your comment 2) about risk of bias and quality of studies will not be addressed.
- The flow diagram of PRISMA has been added as figure Appendix B.
- Selected study data have been added in table Appendix F. I do not think it is relevant to report the IF of the source. Author, year of publishing, the editorial source, the document type, the population or setting, and the findings have been reported.
I would like to thank the Reviewer for the comprehensive evaluation of the article. I believe I have answered the requests. I now hope that the manuscript is worthy of being considered for publication.

This manuscript is a resubmission of an earlier submission. The following is a list of the peer review reports and author responses from that submission.
Round 1
Reviewer 1 Report
Nutritional Orthopedics and Space Nutrition: two 3 sides of the same coin
Matteo Briguglio - Nutrients
The author presents a perspective on the potential benefits of collaborations between nutritional management plans for Orthopedic patients and Space travelers alike. Given the plethora of knowledge that exists on the degenerative effects of microgravity on the musculoskeletal system, it is quite logical that the two above-mentioned groups of individuals could benefit from research applied broadly to Orthopedic care and space travel. A perspective of this kind is thus warranted and intriguing. However, there are substantive concerns with the current organization of the written perspective that make it seem disorganized and difficult to follow. The primary suggestion by Dr. Briguglio is that these two fields share much in common. However, the article is written in a format that isolates the two fields (i.e. section 2 is about nutritional orthopedics and section 3 is about space nutrition). It is better for the assertion of the perspective to address the overlap between the two fields for each stage (i.e. section 2 could be “Pre-operative nutritional program vs. pre-launch nutritional program”; and a later section could be “Nutritional resilience in orthopedic patients vs. nutritional resilience in space”). Though this article could develop into an insightful perspective if carefully reorganized, it lacks clarity in its present form. English grammar is well adhered to overall, but close attention should be paid to ensure that grammar errors are eliminated for further clarity. More specific remarks for each section are listed below.
Abstract
Line 20-21: There is a repeated suggestion throughout this manuscript that is mentioned here in the abstract, which is that the fields of orthopedics and space nutrition are “never crossing”. This is certifiably false. A large body of literature discusses the necessity of appropriate nutrition during space travel and the consequences thereof on the musculoskeletal system (including several references cited in this article [References 107-110, e.g.]). It is recommended that this manuscript be rewritten from a perspective of highlighting the importance of nutrition in light of the plethora of literature on the subject, instead of being written from the perspective of providing a bridge between two unrelated fields, which is something that the manuscript certainly does not do.
- The millennia-old nutritional paradigm on Earth
Lines 43-44: 1. “…doctors recommended diverse dietary supplements for enhancing recovery.” – Please elaborate a little as to what these supplements are.
- Nutritional Orthopedics
Lines 78-79: It is not clear here what Dr. Briguglio means by “it is often preferred to ‘treating all instead of mass screening and then treating few’. “all” and “few” what? Patients, or outcomes associated with malnutrition? Also, why is this statement in quotations? Is it from another source? If so, please cite the source.
Lines 97-101: Fasting before major surgeries is still in full practice in many instances, especially if the surgery involves the gastrointestinal system. These sentences seem to indicate that this practice is no longer widely used. The literature cited (reference 20) alludes to when the practice is used and where it may not be necessary. Please clarify this by elaborating on when fasting is medically considered to be necessary, and when it is not.
- Space Nutrition
Lines 260-265: These sentences are somewhat unclear. Is line 264 supposed to read “light-minutes”, thus alluding to a measure of distance. Also, this seems to be a segue into the following section discussing the importance of establishing the ability to grow food in space. Please rewrite these sentences for clarity and to make the transition smoother by citing references that have discussed the importance of sustainable food growth systems during extended space travel.
Line 270: Is there a reference discussing “Transit Food Systems”?
Line 282: What consequences specifically does galactic radiation cause?
Line 326-327 (& line 374): “The further outside the Milky Way” suggests that the astronauts would be leaving the galaxy. This is dramatically unrealistic for the present era. Please rephrase to reflect our present pursuits (e.g., “the further away from earth” or “the closer to alien worlds”, etc.).
Reviewer 2 Report
Despite this manuscript was not a systematic review, this manuscript comprised a narrative review which did not follow the recommended PRISMA requirements.
According to these criteria, the flow diagram was not provided and the checklist was not followed.
Indeed, searching startegy, inclusion and exclusion criteria, possible bias... were not detailed.
This manuscript could be of interest as a systematic review or following all PRISMA criteria, but this manuscript is not suitable for publication in its current form.